# Cold Resistance of *Euonymus japonicus* Beihaidao Leaves and Its Chloroplast Genome Structure and Comparison with Celastraceae Species

**DOI:** 10.3390/plants11192449

**Published:** 2022-09-20

**Authors:** Hongyu Cai, Xiaozheng Gu, Yongtan Li, Yachao Ren, Shufang Yan, Minsheng Yang

**Affiliations:** 1Forest Department, College of Forestry, Hebei Agricultural University, Baoding 071000, China; 2Hebei Key Laboratory for Tree Genetic Resources and Forest Protection, Baoding 071000, China; 3Hebei Academy of Forestry and Grassland Science, Shijiazhuang 050050, China

**Keywords:** *Euonymus japonicus* Beihaidao, chloroplast genome, *Euonymus*, high variable sites, positive selection, cold resistance

## Abstract

*Euonymus japonicus* Beihaidao is one of the most economically important ornamental species of the *Euonymus* genus. There are approximately 97 genera and 1194 species of plants worldwide in this family (Celastraceae). Using *E. japonicus* Beihaidao, we conducted a preliminary study of the cold resistance of this species, evaluated its performance during winter, assembled and annotated its chloroplast genome, and performed a series of analyses to investigate its gene structure GC content, sequence alignment, and nucleic acid diversity. Our objectives were to understand the evolutionary relationships of the genus and to identify positive selection genes that may be related to adaptations to environmental change. The results indicated that *E. japonicus* Beihaidao leaves have certain cold resistance and can maintain their viability during wintering. Moreover, the chloroplast genome of *E. japonicus* Beihaidao is a typical double-linked ring tetrad structure, which is similar to that of the other four *Euonymus* species, *E. hamiltonianus*, *E. phellomanus*, *E. schensianus*, and *E. szechuanensis*, in terms of gene structure, gene species, gene number, and GC content. Compared to other Celastraceae species, the variation in the chloroplast genome sequence was lower, and the gene structure was more stable. The phylogenetic relationships of 37 species inferred that members of the Euonymus genus do not form a clade and that *E. japonicus* Beihaidao is closely related to *E. japonicus* and *E. fortunei*. A total of 11 functional positive selected genes were identified, which may have played an important role in the process of Celastraceae species adapting to environmental changes. Our study provides important genetic information to support further investigations into the phylogenetic development and adaptive evolution of Celastraceae species.

## 1. Introduction

The family Celastraceae contains approximately 97 genera and 1194 species of plant worldwide, primarily throughout the tropics and subtropics, with some species in temperate regions, and many of which have great economic value [1,2,3]. *Euonymus japonicus* Beihaidao is a cultivated variety of *E. japonicus* Thunb. that was introduced to China from Japan in 1986. Euonymus japonicus Beihaidao is cultivated in urban landscaping in all provinces and regions in north and south China. The leaves are broad, evergreen, and delicate. In autumn, the mature fruits crack and reveal a red aril, which enhances its ornamental value. This species exhibits a high resistance and strong ability to absorb harmful gases; therefore, it serves both ornamental and environmental protection purposes. Therefore, it is important to gather additional information regarding the development and use of *E. japonicus* Beihaidao.

Chlorophyll is the main pigment involved in the process of photosynthesis and plays a major role during the light absorption stage. Its content directly affects the efficiency of photosynthesis. After plants are subjected to low-temperature stress and once this stress intensifies, the morphological structure and function of chloroplasts are damaged, resulting in the reduction of chlorophyll content and the inhibition of photosynthesis. Studies have documented that under low-temperature stress, plant chlorophyll synthesis is weakened and degradation is accelerated, and chlorophyll content continues to decrease with further increases in stress [4,5]. Previous studies showed that low-temperature damages chloroplast ultrastructure and causes chloroplast dysfunction [6,7]. Plant varieties with strong cold resistance will adjust the structure and function of chloroplasts during cold acclimation to ensure the stability of the photosynthetic rate and to adapt to the low-temperature environment [8]. In studies of cold resistance in camphor (*Cinnamommum camphora*), rice (*Oryza sativa* L.), and *Michelia* L., chlorophyll content within varieties with strong cold resistance decreased less under low-temperature stress compared to non-cold resistant varieties [9,10,11].

Under low temperatures, the expression of plant genes is induced or inhibited, which is the molecular mechanism by which plants cope with low-temperature stress. Photosynthetic system genes, genetic system genes, and biosynthetic genes are encoded by the chloroplast genome. Such functional genes play an important role in plant adaptation to environmental changes. When subjected to low-temperature stress, the expression of these genes will change to a certain extent in response. Chloroplast ATP synthase is a key enzyme in the process of oxidative phosphorylation in plants, and it plays an important function in the plant response to abiotic stress. In a previous study [12], the expressions of chloroplast proteins and related genes in sugarcane were affected by low-temperature stress. Low temperature inhibited the *atpC* gene encoding ATP synthase, indicating that the electron transmission and energy synthesis links of sugarcane were destroyed under a low-temperature environment. In addition, the expression of the *PsbO* and *PsbP* genes encoding the photosystem II subunit increased, indicating the complexity of the sugarcane response to low-temperature stress. A study of the molecular mechanism of cold resistance in *Brassica rapa* L. demonstrated that under a low-temperature environment, the expression of the PSBR gene encoding the photosystem II subunit was downregulated to enhance the degree of photoinhibition in response to low-temperature stress [13]. Through the comparative analysis of chloroplast genomes of Actinidia jujuba and three other sequenced Actinidia species, another study found that the differentially expressed *rbcL* and *rpoA* genes under low temperature were candidate genes for cold resistance analysis, which would provide genomic information for the study of cold resistance of related genes in *Actinidia* chloroplasts [14]. Together, these studies show that the chloroplast genome can provide a great deal of genetic information as a basis for the study of plant cold resistance in plants.

Chloroplasts are commonly found in land plants, algae, and some protozoa, and are important organelles with autonomous genetic information within cells [15,16,17]. Similar to mitochondria, they are semi-autonomous organelles with their own independent genome, called the “chloroplast genome” [18,19,20]. The chloroplast genome of most plants is highly conserved, with a typical covalently closed double-stranded loop structure, consisting of a pair of inverted repeat regions (IR), a small single copy region (SSC), and a large single copy region (LSC). The two IR regions of equal length and opposite orientation are separated by the LSC and SSC [21,22]. Very few plants contain linear chloroplast molecules, although maize and *Acetabularia* are exceptions [23,24]. Although the structure of the chloroplast genome is relatively conserved [25,26], it also experiences mutations, including loss of genes as well as genomic structural variants such as insertions/deletions, short fragment inversions, duplications, and gene structural rearrangements [27,28,29]. These variants are of great importance for the study of the origin and evolution of plants and phylogeny [30]. Compared to the nuclear genome, the chloroplast genome is relatively simple in structure and easy to extract full sequences; moreover, its moderate nucleotide substitution rate, conserved gene composition, and stable structure make it suitable for phylogenetic studies among plant taxa [31]. In addition, chloroplast genomes contain a large amount of genetic information, and photosynthesis-related genes and biosynthetic genes are of great significance in plant resistance to environmental changes among them.

Since the first complete chloroplast genome of tobacco was determined [32], the chloroplast genomes of plants have received widespread attention and research; however, studies of the chloroplast genomes within the Celastraceae family are lacking. In this study, the chloroplast genome of *E. japonicus* Beihaidao was sequenced, spliced, and annotated, and was compared to the chloroplast genomes of 12 other species of Celastraceae obtained from NCBI. Our objectives were to provide data on the whole chloroplast genome of *E. japonicus* Beihaidao; to compare the characteristics and structural variation in chloroplast genome sequences of species in the genus *Euonymus*; to study simple repetitive sequences, large repetitive sequences, and hotspot regions as candidate sequences for species identification and phylogenetic studies in the family Celastraceae; and to identify positively selected genes as potential genes for the adaptive evolution of the genus *Euonymus*. The goal was to analyze the role of chloroplast genes in cold resistance of *E. japonicus* Beihaidao and provide a basis for in-depth studies on the evolution, genetic diversity, and population structure of *Euonymus*.

## 2. Materials and Methods

### 2.1. Observation of the External Morphology of the Blade

From September 2020 to May 2021, we conducted a field investigation at Lvlong Nursery Farm, Bazhou City, Hebei Province. Using two local common greening species *E. microcarpus* and *E. oblongifolius* as controls, the leaves of these species and of *E. japonicus* Beihaidao were observed and photographed during the winter. The observation periods were September 2020, December 2020, and May 2021. Three trees of each species were selected for observations and photographs. Based on the level of freezing damage, the symptoms of leaf freezing injury were divided into different grades: Grade 0, no symptoms of damage; Grade 1, leaf color was slightly darker, brown spots on the leaves or a small amount of chlorosis on the edge of leaves; Grade 2, leaves were dull with large brown spots on the backs of leaves or about 20% of young leaves had lost green color on edges and tips of leaves; Grade 3, leaves were brown with large areas of infiltrating spots on the backs of leaves or 20–50% of young leaves had lost green color, turned white, and large areas were shrunken; and Grade 4, leaves were dark brown or boiled in a large area with severe water stains and leaf juice exudation. The lethal temperature of the leaves was expressed by the temperature at which the leaves had reached Grade 4.

### 2.2. Physiological Index Determination

#### 2.2.1. Determination of Color Differences on the Leaf Surface

A CR-400 color difference meter (Konica Minolta, Tokoy, Japan) was used to measure the color changes on the leaf surfaces of the three focal species. Three points were randomly selected on each blade to measure L*, A* and b* values. L* represents the degree of brightness, and the larger the value, the brighter the surface. A* represents the degree of red and green; positive and negative values indicate that the tested sample is red and green, respectively. The magnitude of the absolute value represents the depth of red or green. B* represents the degree of yellow and blue; positive and negative values indicate that the tested sample is yellow and blue, respectively. The magnitude of the absolute value represents the depth of yellow or blue. Three fixed trees were selected for each species as three biological replicates, and 30 leaves were collected from each tree for measurements (average values are reported). The average value represents the color characteristics of the leaf surface.

#### 2.2.2. Determination of Photosynthetic Pigment Content

(1)Chlorophyll levels were measured using the absolute ethanol extraction method. A sample of 0.1 g fresh and clean plant leaf was cut into filaments and then placed in a test tube. After adding 10 mL absolute ethanol, the samples were placed in the dark for chlorophyll extraction at room temperature for about 24 h until the leaves were colorless or white (the soaking time can be appropriately shortened or extended according to the color state of the leaves).(2)The specific type of chlorophyll was determined using the colorimetry method. The test solution was poured into a cuvette, while a blank control cuvette contained anhydrous ethanol. Using a spectrophotometer, absorbances were measured at wavelengths of 663, 647, and 470 nm. The content of each pigment was calculated using the following formulas:chl*a* = (12.25 *D663* − 2.79 *D647*) ∗ *V* ∗ 1000 *m,*(1)
chl*b* = (21.50 *D647* − 5. 10 *D663*) ∗ *V* ∗ 1000 *m*,(2)
Carotenoids = (1000 *D470* − 1.82 chl*a* − 85.02 chl*b*) ∗ 198 *V* ∗ 1000 *m*,(3)
where *D647*, *D663*, and *D470* are the absorbances of the sample at 647, 663, and 470 nm, respectively. *V* is the volume of liquid measured, and *m* is the blade mass.

#### 2.2.3. Determination of Chlorophyll Fluorescence Parameters

During a clear, cloudless day each in September 2020, December 2020, and May 2021, leaves were measured using a PEAMK2 portable fluorometer (Hansatech, King’s Lynn, UK) from 10:00 to 11:00 am. Mature leaves at the same position and of the same age were selected. Prior to measurement, the leaves were clamped with leaf clamps for dark adaptation treatment for 15 to 20 min. After dark adaptation, the leaf clamps were opened, and measurements were immediately taken using the fluorometer. The parameters measured were initial fluorescence (*F*_0_), variable fluorescence (*F_V_*), maximum fluorescence (*F_M_*), and *F_v_*/*F_m_*, which represents the maximum photochemical efficiency. 

#### 2.2.4. Determination of Net Photosynthetic Rate

The net photosynthetic rate (PN) of leaves was measured using a li-6400xt photosynthetic instrument (LI-COR, Lincoln, NE, USA).

### 2.3. Sequencing and Analysis of Chloroplast Genome

#### 2.3.1. Plant Material and Genome Sequencing

Fresh leaves of *E. japonicus* Beihaidao were collected from Lvlong Tree Farm, Langfang City, Hebei Province, China, and total DNA was extracted using the CTAB method. After testing for purity, the concentration and quality of the DNA were determined using agarose gel electrophoresis and a nucleic acid and protein analyzer [33], DNA samples were preserved and sent to Shanghai Ling’en Technology Co., Ltd. (Shanghai, China) for chloroplast genome sequencing (Illumina NovaSeq 6000 platform). The complete chloroplast genome sequences of other species were obtained from NCBI (Table 1), including 12 Celastraceae species (*E. fortunei*, *E. hamiltonianus*, *E. japonicus*, *E. maackii*, *E. phellomanus*, *E. schensianus*, *Catha edulis*, *Maytenus guangxiensis*, *Parnassia palustris*, *Parnassia trinervis*, and *Salacia amplifolia*). These sequences were compared to the resulting sequence obtained for the chloroplast genome of *E. japonicus* Beihaidao.

#### 2.3.2. Chloroplast Genome Assembly and Annotated

Quality control of raw reads data obtained from sequencing was performed using Trimmomatic v0.39 [34] to remove low-quality sequences and junctions and to obtain high-quality clean reads. Chloroplast genome assembly was performed using NOVOPlasty v4.2 [35] software. Sequences with sufficiently high coverage depth and long assembly lengths were selected as candidates and then compared to NT libraries to confirm chloroplast scaffolds. Subsequently, sequences were concatenated according to overlap. The assembled sequences were compared to chloroplast reference genome sequences of closely related species using the BLAST program to determine the starting position and orientation, and the possible partitioning structure of the chloroplast (LSC/IR/SSC) to obtain the final chloroplast genome sequence. The chloroplast genome was predicted using GeSeq [36] software for coding proteins, tRNA, and rRNA genes, and then the predicted initial genes were removed redundancy and the first and last gene and exon/intron boundaries were manually corrected to obtain a highly accurate gene set. Finally, the software Chloroplot [37] was used to generate a fully annotated physical map of the *E. japonicus* Beihaidao chloroplast genome.

#### 2.3.3. Repeat Sequence Analysis

SSR locus analysis was performed on the chloroplast genomes of five *Euonymus* species (*E. japonicus* Beihaidao, *E. hamiltonianus*, *E. phellomanus*, *E. schensianus*, *E. szechuanensis*) using the Microsatellite Identification Tool [38] with the following parameters: at least eight single-nucleotide unit repeats; at least five two-nucleotide unit repeats; at least four three-nucleotide unit repeats; at least three four-, five-, and six-nucleotide unit repeats; and a minimum distance between two SSRs of 100 bp. The online tool REPuter [39] was used to analyze the chloroplast genomes of five *Euonymus* species for long repeat sequences with the following parameters: minimal repeat size of 30 bp, and hamming distance of three. The software identified four types of repeats: forward repeat, reverse repeat, complement repeat, and palindromic repeat.

#### 2.3.4. Analysis of Codon Usage

The CUSP program of EMBOSS v6.6.6.0.0 [40] was used to analyze the codon usage of the chloroplast genomes, to calculate RSCU values, and to obtain heat maps of codon usage in 13 *Euonymus* species.

#### 2.3.5. Comparative Analysis of the Chloroplast Genome Sequences of the Celastraceae

The software mVISTA [41] was used to visualize and compare the chloroplast genome structures of 13 Celastraceae family species and to analyze chloroplast genome sequence similarity in the same species. The software DnaSP v5.1 0 [42] was used to analyze the nucleotide diversity (Pi) of the LSC, SSC, and IR regions of the chloroplast genomes of the Celastraceae family species and to mine the chloroplast genomes of Celastraceae family species for highly variable loci with parameters set to: window length of 300 bp and step size of 200 bp. IR boundaries were mapped using IRscope software [43] to compare the IR boundary features of the chloroplast genomes of the Celastraceae family.

#### 2.3.6. Adaptive Evolutionary Analysis

The homologous single-copy gene families of 13 Celastraceae family species were selected and compared using the MAFFT7.453 software [44]. The amino acid sequence comparison results were converted into nucleic acid comparison results using pal2nal v14 [45]. All gene comparisons were combined and merged together. KaKs_Calculator 2.0 [46] was used to calculate the synonymous (Ka) and nonsynonymous (Ks) mutation values of SNP differential genes. Values of Ka/Ks provide a useful method for assessing whether fitness evolution has occurred in protein-coding genes, with values of Ka/Ks > 1 indicating genes subject to positive selection, Ka/Ks = 1 indicating genes evolving neutrally, and Ka/Ks < 1 indicating genes subject to purifying selection [47]. The amino acid sequences were compared with NR, Swiss-Prot, eggnog, KEGG, and GO databases by BLAST to obtain functional annotation information of the coding genes.

According to the existing three groups of transcriptome data which were sampled in September 2020, December 2020, and May 2021 and sequenced, the transcribed genes were detected for KEGG enrichment analysis, and a heat map was drawn according to the expression levels in different periods.

#### 2.3.7. Phylogenetic Analysis

A total of 37 chloroplast whole genome sequences of other species were downloaded from NCBI (12 Celastraceae, 1 Tapisciaceae, 3 Aquifoliaceae, 1 Pentaphylacaceae, 4 Droseraceae, 10 Saxifragaceae, 4 Hydrangeaceae, 1 Iteaceae, and 1 Penthoraceae; Table 1) and then used to conduct phylogenetic analysis with *E. japonicus* Beihaidao, with *Arabidopsis thaliana* was selected as the outgroup. Protein-coding genes common to all species were selected, and multiple sequences were compared using MUSCLE v3.8.31 [48] software. Phylogenetic trees were constructed using the maximum likelihood (ML) method with the PhyML v3.0 software [49]. The bootstrap was set to 1000 replicates.

## 3. Results

### 3.1. Performance of Freezing Injury of E. japonicus Beihaidao during Overwintering

#### 3.1.1. Comparison of Leaf Freezing Damage Characteristics

The three *Euonymus* species suffered from different degrees of freezing damage during the overwintering period, and the freezing damage at different stages of low temperature also differed (Table 2). The overwintering morphology of various tree species is shown in Figure 1. Plants of *E. microcarpus* died due to low temperature and failed to overwinter successfully. In December 2020, the level of freezing damage to the leaves of *E. japonicus* Beihaidao and *E. oblongifolius* was grade 1, while the level of damage to *E. microcarpus* was grade 3. The leaves of *E. japonicus* Beihaidao were darker and accompanied by a small number of brown spots (Figure 1D). The leaves of *E. oblongifolius* and *E. microcarpus* were darker and chlorotic, and the leaves of the latter had lost water, had yellow spots, and had shrunken (Figure 1E,F). In January 2021, the level of freezing damage to the leaves of *E. japonicus* Beihaidao was grade 2, while that of *E. oblongifolius* and *E. microcarpus* was grade 3. The leaves of *E. japonicus* Beihaidao were dark in color, brown on the leaf surface, but without obvious infiltration spots (Figure 1G). The entire leaves of *E. oblongifolius* and *E. microcarpus* were chlorotic and white, and those of the latter were curled with brown spots (Figure 1H,I). Judging from the overwintering morphology of these three *Euonymus* species, *E. japonicus* Beihaidao exhibited good cold resistance.

#### 3.1.2. Comparison of Anatomical Structures of Leaves

After sampling in September 2020, December 2020, and May 2021, the anatomical structures of the leaves of the three *Euonymus* species were observed using the hand-sectioning method. The slices were placed under an optical microscope to measure the thickness of each type of leaf tissue. The structures are shown in Figure 2, and related statistical data on the anatomical structure of leaves are shown in Table 3. The cold tolerance of plants is closely related to the structural compactness of leaves. During the natural overwintering process, the leaf structural compactness (CTR) increased, and the looseness (SR) decreased (Table 3). After the temperature increased in May 2021, CTR decreased, and SR increased. The leaves of the three species were able to resist the cold environment by increasing the CTR of leaf tissue during the overwintering process. The leaves exhibited the typical tissue structure of dicotyledon leaves (Figure 2). The upper and lower epidermis of leaves are composed of monolayer cells and covered with a thick cuticle. The portion near the upper epidermis is composed of palisade tissue, and the cells are regular in shape and arranged neatly and closely. Spongy tissue is close to the lower epidermis, and the cells are irregular and loosely arranged. Under the low-temperature environment in December, the number of cell layers of palisade tissue increased in *E. japonicus* Beihaidao leaves, from three layers in September to five layers. The cell shape also changed, the length decreased, and the cells became oval (Figure 2A,B). The palisade tissue cells in May were composed of three layers (Figure 2C), and the shape was similar to that in September. The palisade tissue cells of the leaves of *E. oblongifolius* in September and December were composed of three layers (Figure 2D,E), while the palisade tissue cells in May were two-layered (Figure 2F). The palisade tissue cells of *E. microcarpus* in September and December were composed of two layers (Figure 2G,H), and no differences in cell shape were observed. The cells of spongy tissue of the three *Euonymus* species were arranged more closely in December than during the other 2 months, and the cell gap became smaller.

### 3.2. Changes in Physiological Indexes of E. japonicus Beihaidao during Overwintering

#### 3.2.1. Changes in Color Difference of the Leaf Surface

After plants are subjected to low-temperature stress, the pigment content of leaves changes, which directly affects their color. Our results indicate that color changes in the upper and lower surfaces of the leaves occurred in all three focal *Euonymus* species (Figure 3). Under the low-temperature environment in December, the lightness (L*) value, the absolute value of red-green saturation (a*; original value was negative), and the absolute value of yellow-blue saturation (b*; original value was positive) of the upper and lower surfaces of the leaves of the three tree species all tended to decrease. These patterns indicate that the brightness of the upper and lower surfaces of the leaves decreased, and the degrees of green and yellow become lighter due to low temperatures. The upper surface L* of the leaves of the three *Euonymus* species differed significantly in each month (Figure 3A), but values of lower surface L* did not significantly differ in December (Figure 3B). No significant differences in color were observed between *E. oblongifolius* and *E. microcarpus*; however, *E*. *japonicus* Beihaidao differed from both species, indicating that this species exhibited the deepest degree of green color in December (Figure 3C,D). In December, the degree of yellow on the upper surface was the deepest in *E. microcarpus*, while the degree of yellow on the lower surface was the deepest in *E. japonicus* Beihaidao (Figure 3E,F). The upper surface shows that there is no significant difference between *E. japonicus* Beihaidao and *E. oblongifolius*, but both species significantly differed from *E. microcarpus*. The color of the lower surface did not significantly differ between *E. japonicus* Beihaidao and *E. microcarpus*, but the latter was significantly different from *E. oblongifolius*.

#### 3.2.2. Changes in Photosynthetic Pigment Content

After low-temperature stress, the ultrastructure of chloroplasts can be destroyed, resulting in chlorophyll degradation and chlorophyll content reduction. Our findings indicate that the photosynthetic pigment content of leaves of *E**. japonicus* Beihaidao and *E**. oblongifolius* tended to initially decrease and then increase from September to December to May (Figure 4). Pigment content in *E**. microcarpus* also tended to decrease from September to December. Compared to September, the decrease in pigment content in December was largest in *E**. microcarpus* and smallest in *E**. japonicus* Beihaidao, indicating that the chlorophyll degradation of *E**. japonicus* Beihaidao leaves was the least affected by low temperature. According to the level of degradation of chlorophyll content of the three *Euonymus* species under low-temperature stress, the relative resistance of the three *Euonymus* species to low temperature is: *E. japonicus* Beihaidao > *E. oblongifolius* > *E. microcarpus*.

#### 3.2.3. Changes in Chlorophyll Fluorescence Parameters

Values of *F*_0_, *F_v_*, *F_m_*, and *F_v_*/*F_m_* of the three tree *Euonymus* species tended to initially decrease and then increase during the observation period (Figure 5). Values of both *F*_0_ and *F_M_* in December were highest in *E. japonicus* Beihaidao, followed by *E**. oblongifolius*, with the lowest values occurring in *E*. *microcarpus* (Figure 5A,B). Values of *F_V_* in December were highest in *E**. japonicus* Beihaidao, followed by *E**. microcarpus*, with the lowest values occurring in *E*. *oblongifolius* (Figure 5C). The decreasing range of *F_v_/F_m_* from September to December was highest in *E**. oblongifolius*, followed by *E**. microcarpus*, with the lowest values occurring in *E*. *japonicus* Beihaidao (Figure 5D). In December, values of *F_v_/F_m_* did not significantly differ between *E**. oblongifolius* and *E**. microcarpus*, while values for both species significantly differed from those of *E*. *japonicus* Beihaidao. These results indicate that the impact of low-temperature environments on the PS II reaction centers of *E*. *oblongifolius* and *E**. microcarpus* was greater than for *E*. *japonicus* Beihaidao. The degree of damage to the PS II reaction center of *E. japonicus* Beihaidao was the lowest of the three species under a low-temperature environment. Data from May 2021 showed that *F*_0_, *F_v_*, *F_m_*, and *F_v_/F_m_* gradually increased after overwintering, and the four fluorescence parameters of *E**. japonicus* Beihaidao and *E**. oblongifolius* increased; however, values did not return to levels observed prior to overwintering, indicating that with the increase in temperature, the photosynthetic system of the damaged leaves of the two tree species gradually recovered. Changes in the four fluorescence parameters indicate that *E**. japonicus* Beihaidao exhibits the strongest cold resistance to some extent.

#### 3.2.4. Changes in Net Photosynthetic Rate

The net photosynthetic rate of the three *Euonymus* species was lowest in December, with the highest values occurring in *E**. japonicus* Beihaidao followed by *E**. oblongifolius*, with the lowest values occurring in *E**. microcarpus* (Figure 6). After low-temperature stress, the chloroplast structure and thus the photosynthetic organs of the three species may have been damaged, resulting in the decline of photosynthetic capacity. With the gradual increase in temperature in the spring, the photosynthetic capacity of *E**. japonicus* Beihaidao and *E**. oblongifolius* gradually recovered; the net photosynthetic rate increased, and the photosynthetic capacity was higher than levels prior to overwintering. This response was more pronounced in *E**. japonicus* Beihaidao than in *E**. oblongifolius*. The results of variance analysis indicated that the net photosynthetic rate did not significantly differ between *E**. oblongifolius* and *E**. microcarpus* in either September or December. The net photosynthetic rate of *E**. japonicus* Beihaidao was significantly higher than those of *E**. oblongifolius* and *E**. microcarpus* during each month, indicating that the photosynthetic capacity of *E**. japonicus* Beihaidao was highest.

### 3.3. Analysis of E. japonicus Beihaidao Chloroplast Genome

#### 3.3.1. Basic Characteristics of the *E. japonicus* Beihaidao Chloroplast Genome

The sequenced *E**. japonicus* Beihaidao chloroplast genome is a typical double-stranded circular tetrad structure, 157,661 bp in length, including a pair of 26,683 bp IR regions (IRa, IRb), an LSC region of 85,932 bp and an SSC region of 18363 bp; no deletions of large segments of regional bases were detected (Figure 7). A comparison of the basic chloroplast genome characteristics of five *Euonymus* species showed that the total length of the chloroplast genome of the genus ranged from 157,360 to 157,702 bp (Table 4). The lengths of the LSC region, SSC region, and IR region ranged from 85,932–86,399 bp, 18,317–18,536 bp, and 26,322–26,683 bp, respectively. The lengths of the coding and non-coding regions ranged from 78,069–80,871 bp and 76,594–79,633 bp, respectively. The chloroplast genomes of the five *Euonymus* species encoded 130–134 genes, including 85–89 protein-coding genes, 37 tRNAs, and eight rRNAs. The GC content of the chloroplast genomes of the five *Euonymus* species varied according to location and coding genes, primarily as follows: the coding region (38.02–38.19%) was higher than the non-coding region (36.21–36.44%). the IR region had the highest GC content (42.62–42.72%), followed by the LSC region (35.00–35.18%) and the SSC region (31.63–31.78%). The total GC content (37.18–37.30%) was higher than those of the LSC and SSC regions and lower than that of the IR regions. 

#### 3.3.2. IR Boundary Analysis

The IR boundaries of the chloroplast genomes of Celastraceae were visualized and compared to those of 12 *Euonymus* species (Figure 8). The results indicated that eight protein-coding genes, *rpl22*, *rps19*, *rpl2*, *ycf1*, *ndhF*, *rpl32*, *trnN*, and *trnH*, were present at the junctions of LSC/IR (JL) and SSC/IR (JS). According to the IR boundary genes, the family Celastraceae can be divided into the following four types: type I (*E**. japonicus* Beihaidao, *E*. *fortunei*, *E**. schensianus*, *C**. edulis*, *M**. guangxiensis*, *P**. trinervis*), type II (*E**. japonicus* and *M**. guangxiensis)*, type III (*E**. hamiltonianus*, *E*. *maackii*, *E**. phellomanus*, *E**. szechuanensis*), and type IV (*P**. palustris*). The junctions of LSC/IRb (JLB) of type III occurred between *rps19* and *rpl2*, with *rps19* on the left side of the JLB, 11 bp away, and *rpl2* on the right side of the JLB, 45–60 bp away. By contrast, the JLBs of types I, II, and IV occurred between *rps22* and *rpl9*, and the distances were 12–110 bp and 7–90 bp, respectively. The junctions of IRb/SSC (JSB) of 11 of the Celastraceae species were located in *ycf1*; the exception was *P**. trinervis*, for which the JSB was located in *ndhF*. The junction of SSC/IRa (JSA) was located in the *ycf1* gene for all species except *M**. guangxiensis* and *P**. palustris*. The JSAs of these two species were located between *ycf1*–*trnN* and *rps15*–*trnN*, respectively, at 946 bp and 273 bp from *ycf1* and *rps15*, respectively, and 1278–1552 bp from the *trnN* gene. The junctions of IRa/LSC (JLA) in *E**. japonicus* Beihaidao, *E**. japonicus*, *E**. schensianus*, *M**. guangxiensis*, *P**. palustris*, *P**. trinervis*, and *S**. amplifolia* were all to the left of *trnH*, and the *JLA* of the remaining seven species was in *trnH*.

#### 3.3.3. Comparative Sequence Analysis of Repetitive Sequences of *E. japonicus* Beihaidaos and *Genus euonymus*

Due to the high polymorphism rate of SSRs at the species level, they are considered one of the major sources of molecular markers. Such markers have been used extensively in phylogenetic studies and population genetics. A total of 197–209 SSRs were detected in the chloroplast genomes of five *Euonymus* species, the largest number (169–187) of which was a single nucleotide (Figure 9A). In addition, HexaNucl of *E*. *japonicus* Beihaidao was substantially lower in number than the other four *Euonymus* species. The lengths of these SSRs were mainly around 7–9 bp (Figure 9B). Long repetitive sequences greater than 30 bp in length may promote chloroplast genome rearrangements and increase genetic diversity in the species. Long repeat sequences (25–107 in total) were detected in the chloroplast genome of *Euonymus* species, including 8–44 forward repeats, 3–49 reverse repeats, 7–17 palindromic repeats, and 1–7 complementary repeats (Figure 9C). Most long repeats were 30–34 bp in length, and none were 60–64 bp (Figure 9D). In *E*. *japonicus* Beihaidao, reverse repeats and 30–40 bp long repeats were substantially more common than in the other four *Euonymus* species.

#### 3.3.4. Changes in Net Photosynthetic Rate

To clarify the phylogenetic position and evolutionary relationships of *E**. japonicus* Beihaidao in the Celastraceae, the shared protein-coding genes of 38 species, including *E**. japonicus* Beihaidao, were clustered and analyzed with *Arabidopsis* as an outgroup. The results are consistent with the traditional morphological classification of plants (Figure 10). In total, 13 species of Celastraceae were clustered together, with *E**. japonicus* Beihaidao clustered with *E**. japonicus* and *E. fortunei*, all three of which are closer to each other than to other species in the genus *Euonymus*. Two species in the genus *Parnassia* are clustered together, and three species in the genera *Maytenus*, *Catha*, and *Salacia* are clustered together.

#### 3.3.5. Analysis of Differential Loci

Using mVISTA software, multiple sequence comparisons were performed for the 13 Celastraceae species (Figure 11). Three species, *P**. palustris*, *P**. trinervis*, and *S**. amplifolia*, clearly differed from the remaining 10. Furthermore, the level of genetic variation in *Euonymus* was lower than in other Celastraceae species, with the LSC and SSC regions exhibiting more variation than the IR region and the non-coding region being more variable than the coding region. Dnasp v5.10 software was used to mine sites of variation between the chloroplast genomes of Celastraceae species. There were 11 such sites between the genomes of eight species of the genus *Euonymus* (*E**. japonicus* Beihaidao, *E**. maackii, E**. fortunei*, *E. phellomanus*, *E**. szechuanensis*, *E**. hamiltonianus*, *E**. schensianus*, and *E**. japonicus*), all located in the intergenic region, with the highest number of sites in the LSC region (8 sites). The mean Pi values for such sites in the LSC, SSC, and IR regions were 0.054, 0.063, and 0.045, respectively (Figure 12A, Table 5). There were nine such sites between the genomes of the 13 Celastraceae species, with eight in the intergenic region and one in the genic region; most sites were in the LSC region (seven) and no sites were in the IR region. The Pi means of these sites in the LSC and SSC regions were 0.167 and 0.131, respectively (Figure 12B, Table 6). Such sites of variation will prove useful for studying the phylogeny of Celastraceae species.

#### 3.3.6. Adaptive Evolutionary Analysis

##### Screening of Positive Selection Genes

Using *E**. japonicus* Beihaidao as a control, the single-copy CDS genes of 13 Celastraceae species were analyzed for positive selection, and the ratios of synonymous to nonsynonymous mutations (Ka/Ks) were calculated (Table 7). Here, 11 genes had values of Ka/Ks > 1 (*p* < 0.05). For example, the *atpI*, *rpl20*, *rpl2*, and *rps8* genes had Ka/Ks > 1 in *E**. schensianus* vs. *E**. japonicus* Beihaidao and in *E**. szechuanensis* vs. *E**. japonicus* Beihaidao; *atpE* had Ka/Ks > 1 in *E**. hamiltonianus* vs. *E**. japonicus* Beihaidao and in *E**. szechuanensis* vs. *E**. japonicus* Beihaidao; and *ndhB* had Ka/Ks > 1 in *S. amplifolia* vs. *E**. japonicus* Beihaidao, indicating that these genes have been subject to positive selection during evolution.

##### Positive Selection Genes Annotated and Related Important Metabolic Pathways Analysis

The sequences of the 11 positive selection genes were compared by using NR, Swiss-Port, COG-eggNOG, KEGG, and GO databases (Table 8). The results showed that *atpE*, *atpI*, *ndhB*, and *ndhC* were related to energy production and conversion, and *rpl2*, *rpl20*, *rps8*, and *rps12* are encoding genes of ribosomal proteins, which were related to translation, ribosomal structure, and biosynthesis. AtpE and atpI are the ATP synthase subunits of CF1 epsilon and CF0 A, respectively. NdhB and ndhC belong to the NADH dehydrogenase subunit. PetL is a cytochrome b6/f complex. PsaI is a photosystem I reaction center subunit and psbH is a photosystem II reaction center protein (Table 8 and Table 9).

As shown in Figure 13 and Figure 14, *ndhB*, *ndhC*, *atpE*, *atpI* were annotated to the Oxidative phosphorylation pathway, and have two main functions: 1. catalyzing the dehydrogenation of NADH to release electrons to generate NAD^+^, 2. catalyzing the binding of ADP with Pi to generate ATP. *AtpE*, *atpI*, *psaI*, *psbH* were annotated to the Photosynthesis pathway which can promote ATP production. These two pathways are associated with the metabolism, photosynthesis, and energy supply of *E. japonicus* Beihaidao, which is critical in cold conditions.

##### Expression Analysis of Positive Selection Genes in Different Periods

Four matching results were obtained by searching the positive selection genes in the transcriptome, *rpl2-A*, *psbH*, *ndhB1*, and *atpI*. KEGG enrichment analysis of these genes showed that *psbH*, *ndhB*, and *atpI* were enriched in the Energy Metabolism pathway, and *rpl2-A* was enriched in the Translation pathway (Figure 15). This result was consistent with the annotation results in the chloroplast genome.

A heat map based on expression levels of target genes at different times in the samples (Figure 16) showed that the expression level of *rpl2-A* was down-regulated in December 2020 (cold conditions), and expressions of *psbH*, *ndhB1*, *atpI* were up-regulated in December 2020. It was speculated that *rpl2* was negatively correlated with the cold resistance of *E. japonicus* Beihaidao, and *psbH*, *ndhB*, *atpI* were positively correlated with the cold resistance of *E. japonicus* Beihaidao.

## 4. Discussion

### 4.1. Responses of Leaf Morphological Structure and Physiological Indexes of E. japonicus Beihaidao to Wintering

Under low-temperature stress, not only the growth and development of plants will be affected, but also the external morphology, internal structure, and physiological—biochemical indexes will change due to continuous low temperature. The leaf of *E. japonicus* Beihaidao became darker, lost water, and shrunk under low temperatures in winter. In this study, the palisade tissue increased from three layers to five layers in winter, and the cell morphology became oval. The palisade tissue was higher than the control plant. After the temperature rebounded, it showed the opposite trend, which was consistent with the research results of Zeng and Gao [50,51], indicating that plants adapted to low temperatures by increasing the proportion of palisade tissue during overwintering. In addition, the variation trend of leaf surface color was consistent with that of photosynthetic content. The fluorescence parameters F_0_, F_v_, F_m_, and F_v_/F_m_ were significantly decreased under low temperatures, and this trend was also found in rape and cassava [52,53]. Under low-temperature stress, the decrease in F_v_/F_m_ value and chlorophyll content of *E. japonicus* Beihaidao was the smallest. These results indicate that the damage degree of *E. japonicus* Beihaidao is low and *E. japonicus* Beihaidao had a certain cold resistance ability.

### 4.2. IR Contraction and Expansion

The observed chloroplast genome of *E. japonicus* Beihaidao in this experiment was highly conserved and exhibited the same structure as those of other Celastraceae species [54]; that is, a typical double-stranded cyclic tetrad structure. The genome had a full-length sequence of 157,661 bp, including a large single-copy region (LSC, 85,932 bp), a small single-copy region (SSC, 18,363 bp), and two inverted repeat regions (IRs, 26,683 bp), encoding a total of 132 genes. IR sequence deletions have been found in plants of the families Taxodiaceae [55], Ranunculaceae [56,57], and Euphorbiaceae, and gene rearrangements have been observed [58]. However, in the present study, no gene deletions or gene structure variation was found in the *E. japonicus* Beihaidao chloroplast genome, suggesting that the structure of its chloroplast genome is relatively stable. GC content plays an important role in genome identification, and changes in base composition can be observed in the genomes of different species. Genomic differences among species are reflected by changes in base composition [59]. The GC content of five Euonymus species ranged from 37.18% to 7.30%, which is within the normal range across the chloroplast genomes of seed plants (34–40%). The highest GC content occurred within the IR region, while the lowest was in the SSC region, mainly because the IR region contained eight GC rRNA genes and 18–19 tRNA genes, both with high GC content. The high GC content in the IR region may explain its stability compared to the LSC and SSC regions.

### 4.3. Genome Sequence Variation

SSRs are molecular markers with high variability within the same species and can be used for population genetic and polymorphism studies [60]. In the present study, a total of 197–209 SSRs were detected in the chloroplast genomes of five species of the genus Euonymus. Most of them were A/T type mononucleotides, which supports the finding that SSRs in chloroplast genomes are usually composed of short polythymine (polyT) or polyadenine (polyA) repeats [61]. These results suggest that mononucleotide nucleotide repeats may play a more important role in gene variation than other types of SSRs. Large and complex repeats may function in the alignment and recombination of chloroplast genomes [62]. For example, a total of 25–107 bp repeats were detected in the chloroplast genomes of five species of the genus Euonymus, and most repeats were between 30–34 bp in size, which is similar to those reported for other plants [63,64]. In the present study, a comparison of chloroplast genome sequences using mVISTA showed high similarity among the chloroplast genomes of 13 Celastraceae species. A large number of highly variable sequence regions were identified using sliding-window analysis, most of which were primarily located in intergenic regions. These regions also exhibited the highest number of variable loci in the LSC region, which is consistent with findings from other taxa [65]. The variation mainly occurred in *P. palustris*, *P. trinervis*, *S. ampholia*, and *C. edulis*, which is consistent with the outcome of the evolutionary tree. The highly variable sequence trnH-psbA, identified in this study, is a common DNA barcode for land plants [66] and has been widely used for identification within the Asteraceae family and genus Fagopyrum [67,68]. In addition, the ycf1 gene has also been reported to have a high variation hotspot region in the *Papaveraceae* family [69]. The high-variation hotspots identified in this study could serve as a potential resource for developing molecular markers for phylogenetic analysis and identification of species in the Celastraceae family. The results of codon usage indicated that the chloroplast genome third codon of 13 Celastraceae species had an A/T base preference, which is similar to that of the Lauraceae [3] and Leguminosae [70] families as well as other plants.

### 4.4. Phylogenetic Evolution

In the present study, the chloroplast genome sequences of 37 species were downloaded from NCBI, and the shared protein-coding genes of 38 species, including those of *E. japonicus* Beihaidao obtained from this study, were clustered and analyzed to construct a phylogenetic tree. The clustering results were consistent with the traditional classification system, in which *E. japonicus* Beihaidao is clustered with *E. japonicus* and *E. fortunei*. In addition, two species of the genus Parnassia clustered together with species of the family Celastraceae, Parnassia was previously classified as Lycoris [71,72], which has been classified as Celastraceae family based on the APG IV classification [73]. In phylogenetic analyses of the chloroplast genomes of P. wightiana [74] and P. palustris [75], the Parnassia clustered together with the Euonymus under the family Celastraceae, our clustering results are consistent with those findings, suggesting that chloroplast genomes can provide a database for studying species phylogeny.

### 4.5. Genetic Adaptive Evolution

Analysis of the adaptive evolution of genes is valuable for studying changes in gene function and structure as well as species evolution [76]. In the present study, selection pressure analysis of homologous single-copy gene families from 13 Celastraceae species showed that most protein-coding genes exhibited values of Ka/Ks < 1, indicating purifying selection. Previous studies have shown that positive selection is less common than neutral evolution or purifying selection [77]. In all, 11 positively selected genes were detected in this study, seven of which were photosynthetic genes and four of which were self-replicating genes. These results are consistent with the conclusion of the previous physiological performance that the cold resistance of the leaves of *E. japonicus* Beihaidao was mainly presented at a higher level of chlorophyll, photosynthetic pigment, and photosynthetic reaction than other species under cold conditions. Photosynthesis can provide energy to maintain plant survival under low-temperature conditions. Chloroplast ATP synthase and NADH dehydrogenase are key enzymes in the process of oxidative phosphorylation of plants and play an important role in plant response to abiotic stress. The atpE, atpI, ndhB, and ndhC detected in this study may play an important role in the adaptation of species to environmental changes within the Celastraceae family. Through the analysis of combining transcriptome, the expression level of rpl2-A was down-regulated in December 2020, and the expression levels of psbH, ndhB1, and atpI were up-regulated in December 2020. It was speculated that rpl2 was negatively correlated with the cold resistance of *E. japonicus* Beihaidao; and psbH, ndhB, and atpI were positively correlated with the cold resistance of *E. japonicus* Beihaidao. Among the 11 positively selected genes detected in this study, petL, rpl20, rps12, and rps8 have all reported to be positively selected [78,79]. Previous studies have shown the rpl2 gene is upregulated under cold stress [80,81]. It was speculated that these genes may be related to the cold resistance of *E. japonicus* Beihaidao. However, the response of plants to low-temperature stress is complex and not only related to chloroplast genes, and more molecular evolutionary biology studies are needed to understand the cold resistance mechanism and adaptative evolution in plastid genes.

## 5. Conclusions

The chloroplast genome of *E. japonicus* Beihaidao is a typical double-stranded cyclic tetrad structure, 157,661 bp in length, that encodes 132 genes with 37.26% GC content. The genome is similar in size, gene type, gene number, and GC content to the chloroplast genomes of the other four focal species of *Euonymus,* with SSRs dominated by (A)n and (T)n. *Euonymus*
*japonicus* Beihaidao shares the A/T preference in the use of codon three with the other 12 Celastraceae species, but the genome is less variable. The 11 positively selected genes identified may have played an important role in the evolution of different species in the family. The phylogenetic analysis clearly demonstrated that *E. japonicus* Beihaidao is more closely related to *E. japonicus* and *E. fortunei*, supporting the position of species of the genus *Euonymus* in the Celastraceae family.

## Figures and Tables

**Figure 1 plants-11-02449-f001:**
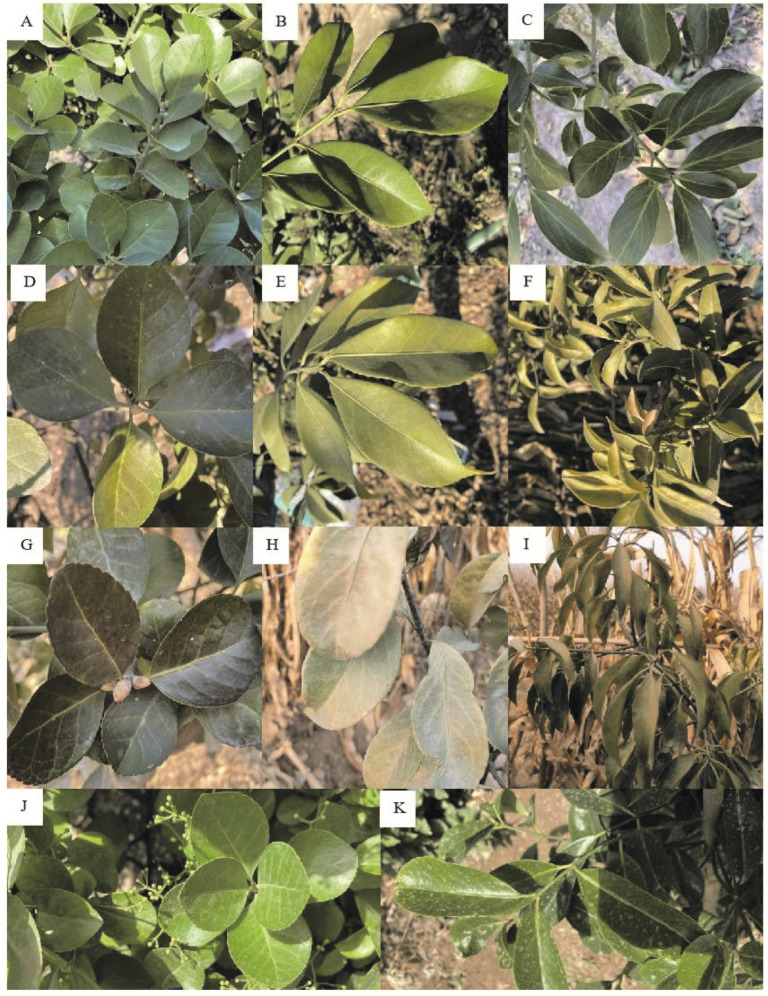
Leaf morphological changes of three species. (**A**) September 2020, *E. japonicus* Beihaidao; (**B**) September 2020, *E. oblongifolius*; (**C**) September 2020, *E. microcarpus*; (**D**) December 2020, *E. japonicus* Beihaidao; (**E**) December 2020, *E. oblongifolius*; (**F**) December 2020, *E. microcarpus*; (**G**) January 2021, *E. japonicus* Beihaidao; (**H**) January 2021, *E. oblongifolius*; (**I**) January 2021, *E. microcarpus*; (**J**) May 2021, *E. japonicus* Beihaidao; (**K**) May 2021, *E. oblongifolius*.

**Figure 2 plants-11-02449-f002:**
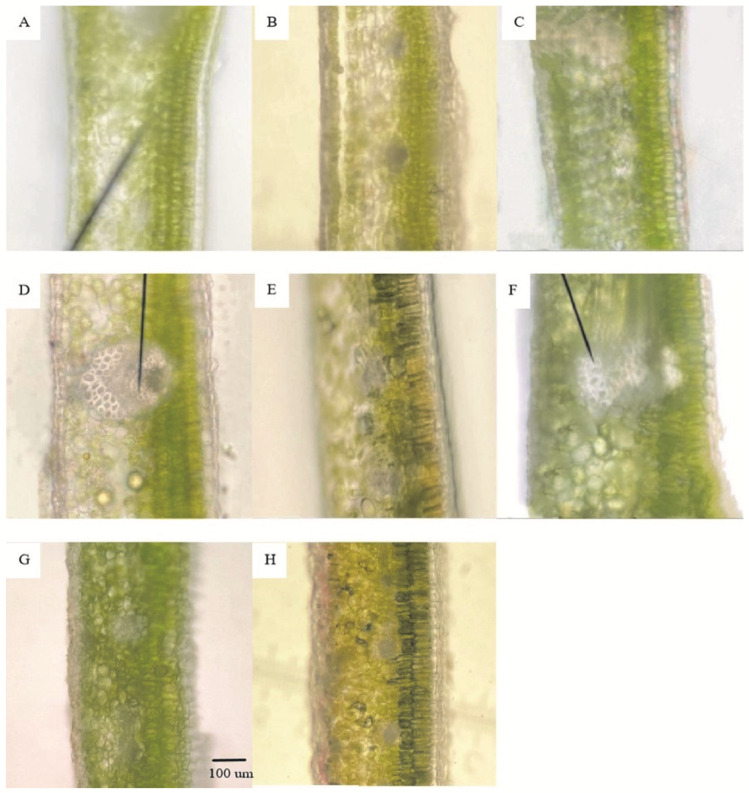
Schematic diagram of leaf anatomical structure. (**A**) September 2020, *E. japonicus* Beihaidao; (**B**) December 2020, *E. japonicus* Beihaidao; (**C**) May 2021, *E. japonicus* Beihaidao; (**D**) September 2020, *E. oblongifolius*; (**E**) December 2020, *E. oblongifolius*; (**F**) May 2021, *E. oblongifolius*; (**G**) September 2020, *E. microcarpus*; (**H**) December 2020, *E. microcarpus*.

**Figure 3 plants-11-02449-f003:**
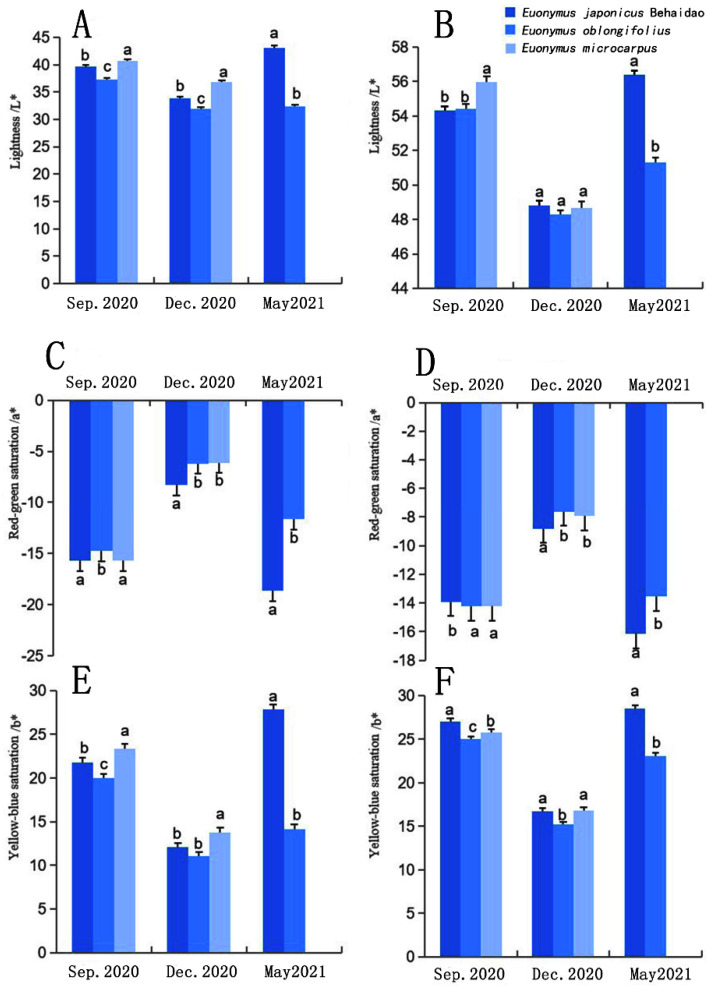
Changes in leaf surface color difference in three *Euonymus* species. (**A**) lightness of the upper surface; (**B**) lightness of the lower surface; (**C**) red-green saturation of the upper surface; (**D**) red-green saturation of the lower surface; (**E**) yellow-blue saturation of the upper surface; (**F**) yellow-blue saturation of the lower surface; a, b and c represented significant differences.

**Figure 4 plants-11-02449-f004:**
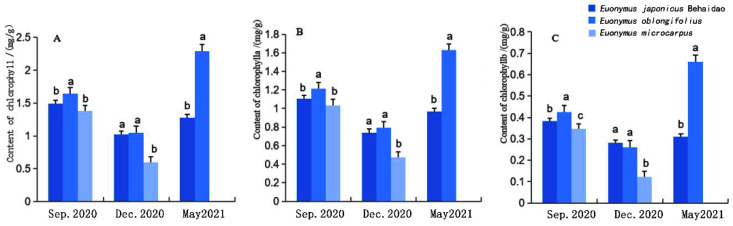
Changes in pigment content in three *Euonymus* species. (**A**) total chlorophyll content; (**B**) chlorophyll *a* content; (**C**) chlorophyll *b* content; a, b and c represent significant differences.

**Figure 5 plants-11-02449-f005:**
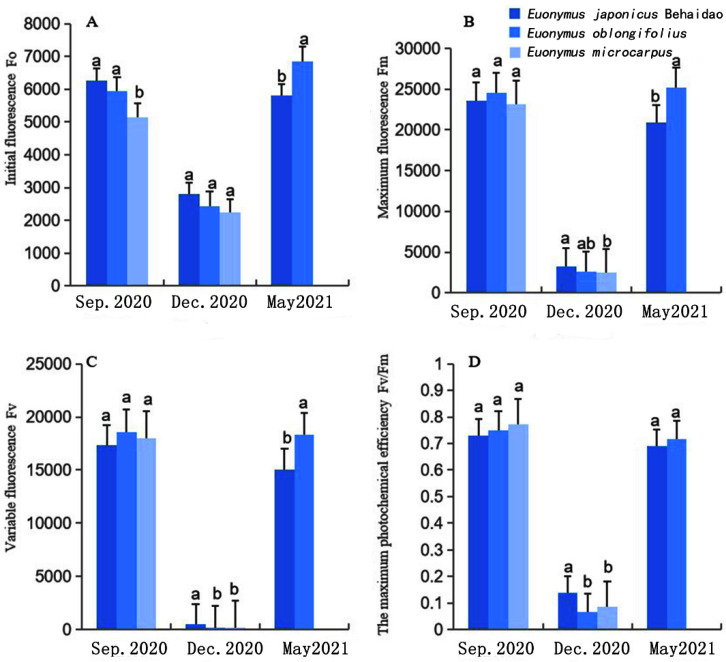
Changes in the chlorophyll fluorescence parameters in three *Euonymus* species. (**A**) initial fluorescence; (**B**) maximum fluorescence; (**C**) variable fluorescence; (**D**) the maximum photochemical efficiency; a and b represent significant differences.

**Figure 6 plants-11-02449-f006:**
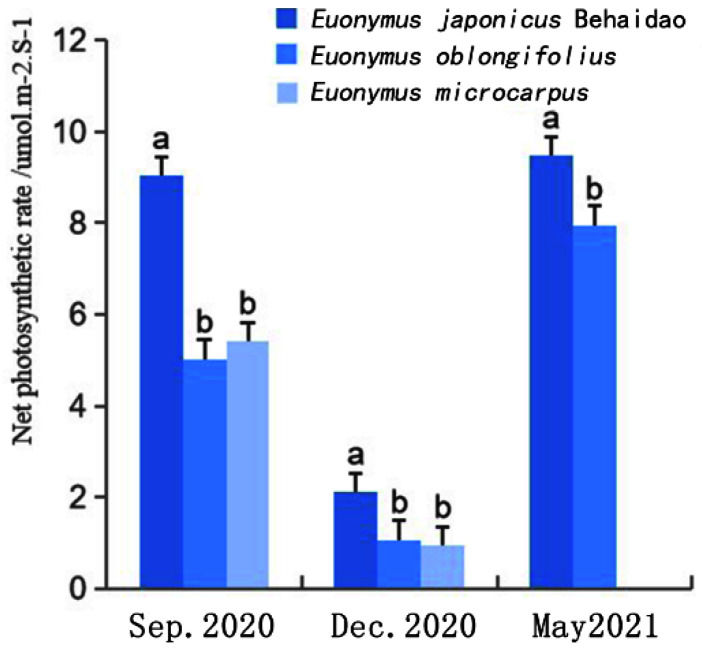
Changes of the net photosynthetic rate in three species; a and b represent significant differences.

**Figure 7 plants-11-02449-f007:**
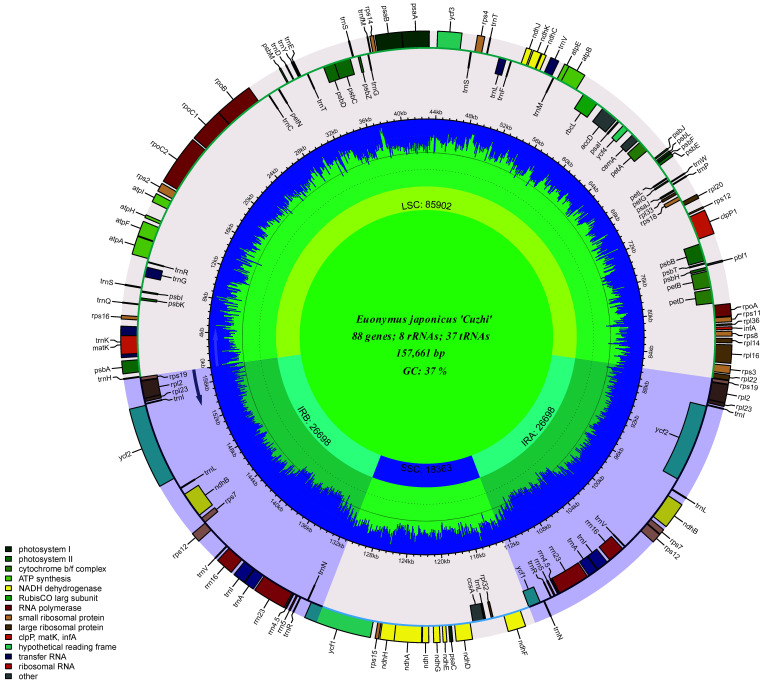
Physical map of the chloroplast genome of *E. japonicus* Beihaidao.

**Figure 8 plants-11-02449-f008:**
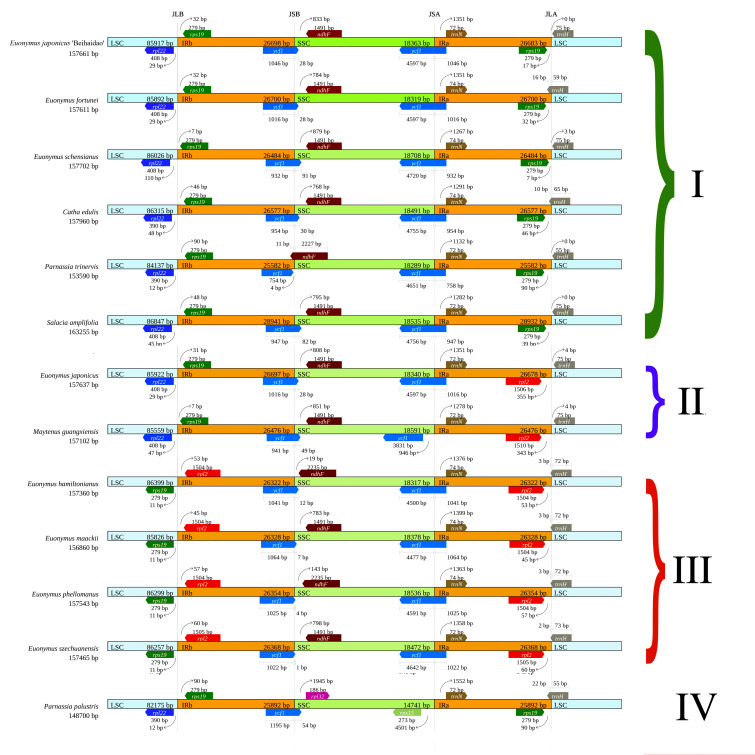
IR boundary figure. According to the IR boundary genes, the family Celastraceae can be divided into the following four types: type I *(E. japonicus Beihaidao, E. fortunei, E. schensianus, C. edulis, M. guangxiensis, P. trinervis*), type II (*E. japonicus and M. guangxiensis*), type III (*E. hamilto-nianus, E. maackii, E. phellomanus, E. szechuanensis*), and type IV (*P. palustris*).

**Figure 9 plants-11-02449-f009:**
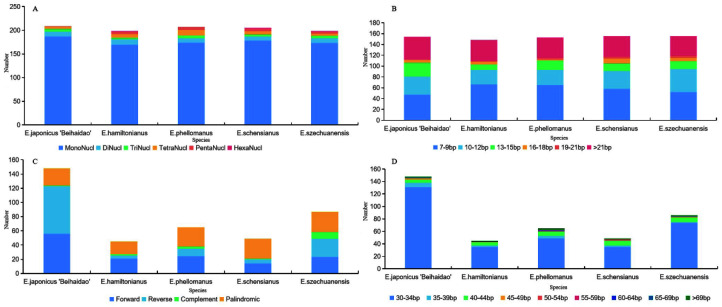
Repeat sequence statistics of *Euonymus*. (**A**) Type of SSR; (**B**) Length distribution of SSR; (**C**) Type of long repeats; (**D**) Length distribution of long repeats.

**Figure 10 plants-11-02449-f010:**
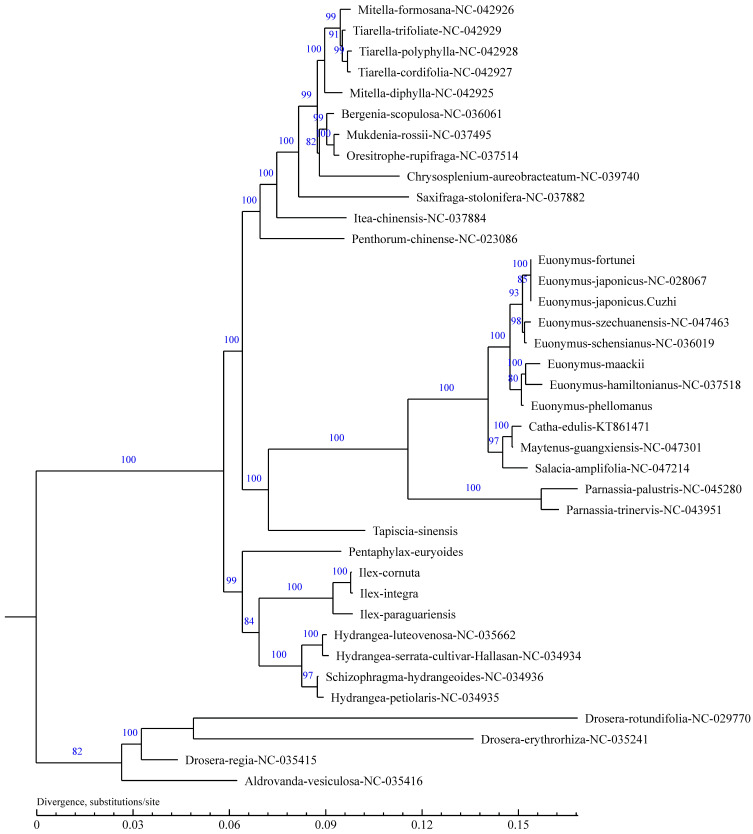
Phylogenetic tree based on complete chloroplast genome sequences from 38 species. The numbers on the branches represent reliability, and the maximum is 100.

**Figure 11 plants-11-02449-f011:**
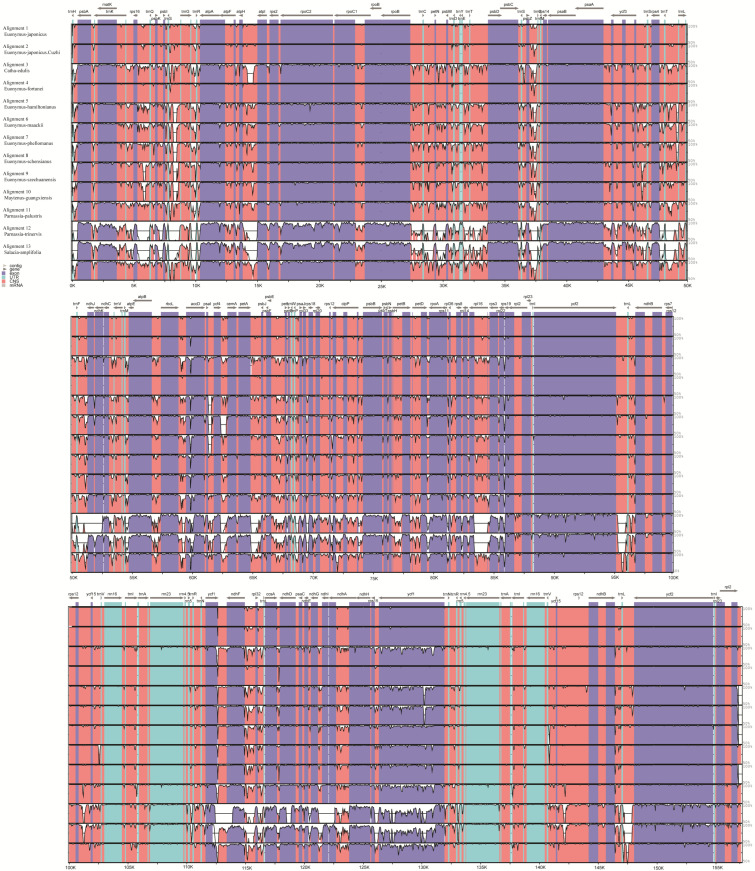
Comparison of chloroplast genome sequences among 13 Celastraceae species.

**Figure 12 plants-11-02449-f012:**
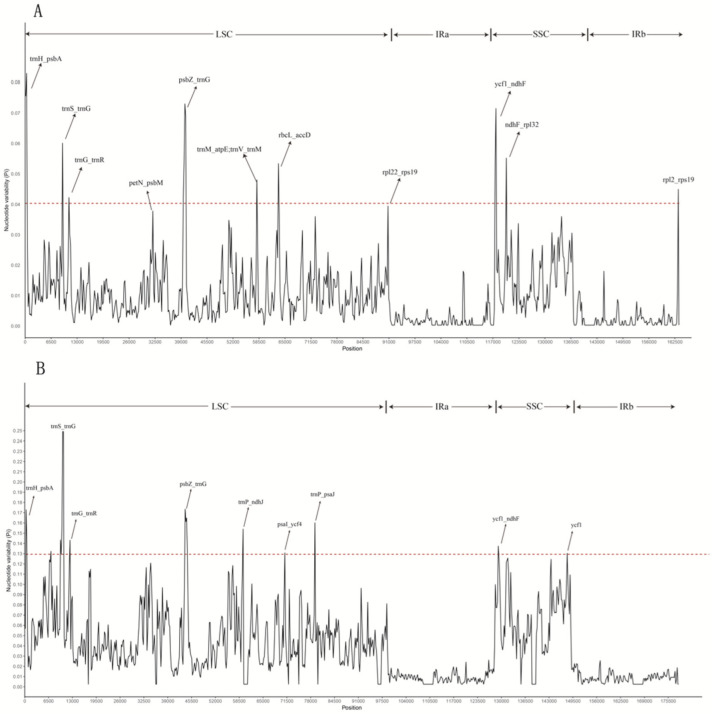
Sliding-window analysis of chloroplast genome sequences of (**A**) *Euonymus* species and (**B**) Celastraceae species.

**Figure 13 plants-11-02449-f013:**
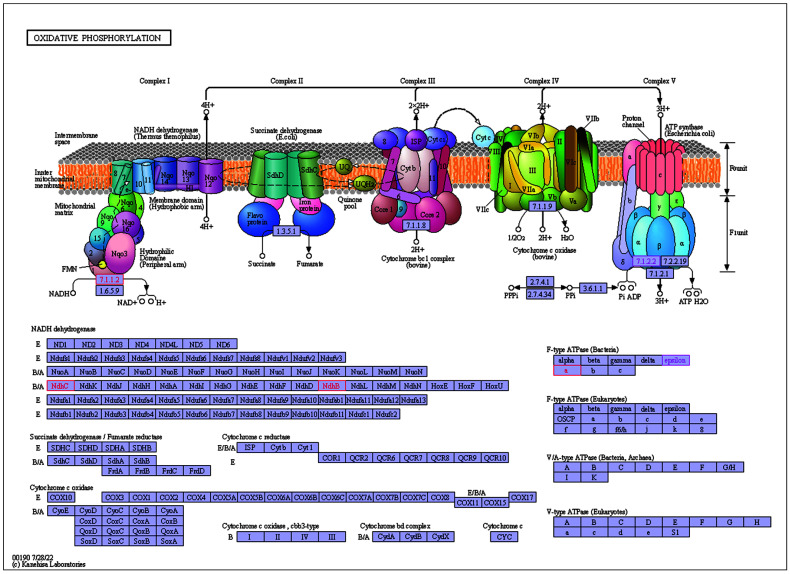
Details of Oxidative phosphorylation pathway.

**Figure 14 plants-11-02449-f014:**
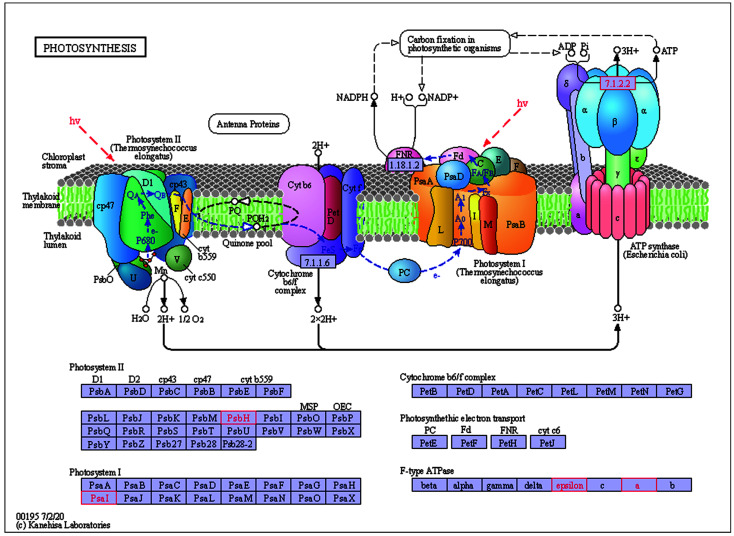
Details of Photosynthesis pathway.

**Figure 15 plants-11-02449-f015:**
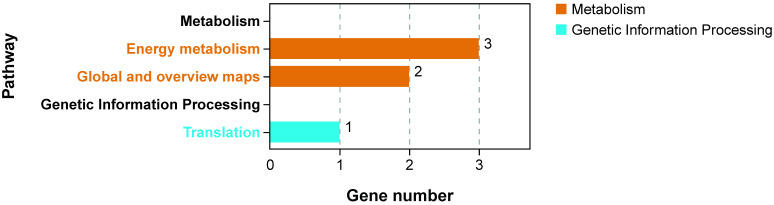
KEGG enrichment analysis of positive selection genes.

**Figure 16 plants-11-02449-f016:**
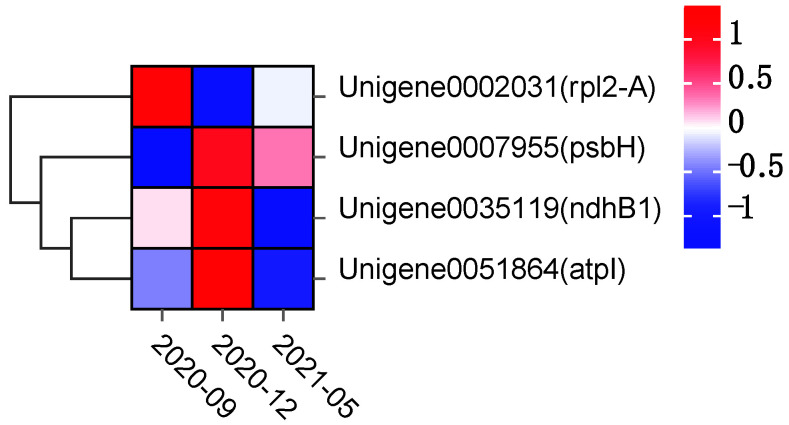
Heatmap of Gene Expression at Different Stages.

**Table 1 plants-11-02449-t001:** Species used for phylogenetic analysis and their NCBI accession numbers.

Serial Number	Species	Registry Number Accession	Serial Number	Species	Registry Number Accession
1	*Euonymus maackii*	NC_057059	20	*Drosera regia*	NC_035415
2	*Euonymus fortunei*	NC_057058	21	*Drosera rotundifolia*	NC_029770
3	*Euonymus phellomanus*	NC_057060	22	*Saxifraga stolonifera*	NC_037882
4	*Euonymus szechuanensis*	NC_047463	23	*Mitella diphylla*	NC_042925
5	*Euonymus hamiltonianus*	NC_037518	24	*Mitella formosana*	NC_042926
6	*Euonymus schensianus*	NC_036019	25	*Mukdenia rossii*	NC_037495
7	*Euonymus japonicus*	NC_028067	26	*Bergenia scopulosa*	NC_036061
8	*Maytenus guangxiensis*	NC_047301	27	*Tiarella cordifolia*	NC_042927
9	*Salacia amplifolia*	NC_047214	28	*Oresitrophe rupifraga*	NC_037514
10	*Parnassia palustris*	NC_045280	29	*Tiarella polyphylla*	NC_042928
11	*Parnassia trinervis*	NC_043951	30	*Tiarella trifoliata*	NC_042929
12	*Catha edulis*	KT861471	31	*Chrysosplenium aureobracteatum*	NC_039740
13	*Ilex paraguariensis*	NC_031207	32	*Hydrangea luteovenosa*	NC_035662
14	*Ilex cornuta*	NC_044416	33	*Hydrangea petiolaris*	NC_034935
15	*Ilex integra*	NC_044417	34	*Hydrangea serrata*	NC_034934
16	*Pentaphylax euryoides*	NC_035710	35	*Schizophragma hydrangeoides*	NC_034936
17	*Tapiscia sinensis*	NC_036960	36	*Itea chinensis*	NC_037884
18	*Aldrovanda vesiculosa*	NC_035416	37	*Penthorum chinense*	NC_023086
19	*Drosera erythrorhiza*	NC_035241			

**Table 2 plants-11-02449-t002:** Frost damage levels of three *Euonymus* species.

Species	Frost Damage Level	
	December 2020	January 2021
*E. japonicus* Beihaidao	Grade 1	Grade 2
*E. oblongifolius*	Grade 1	Grade 3
*E. microcarpus*	Grade 3	Grade 3

**Table 3 plants-11-02449-t003:** Changes in leaf tissue structure compactness of three *Euonymus* species.

Sample	Species	Time	Thickness of Palisade Tissue	Thickness of Spongy Tissue	CTR	SR
1	*E. japonicus* Beihaidao	2020.09	116.42 ± 9.17 ^a^	153.05 ± 24.96 ^a^	36.89 ± 6.20 ^a^	48.02 ± 7.50 ^a^
2	*E. japonicus* Beihaidao	2020.12	90.00 ± 10.00 ^b^	105.00 ± 5.00 ^b^	38.52 ± 1.48 ^a^	45.37 ± 4.63 ^a^
3	*E. japonicus* Beihaidao	2021.05	103.66 ± 19.24 ^a^	139.04 ± 17.76 ^a^	34.28 ± 4.94 ^a^	46.36 ± 6.91 ^a^
4	*E. oblongifolius*	2020.09	106.66 ± 5.00 ^a^	130.00 ± 15.00 ^a^	40.95 ± 3.56 ^a^	49.82 ± 0.81 ^a^
5	*E. oblongifolius*	2020.12	146.33 ± 31.14 ^a^	166.25 ± 20.75 ^a^	41.46 ± 5.44 ^a^	47.43 ± 3.13 ^a^
6	*E. oblongifolius*	2021.05	115.71 ± 52.61 ^a^	141.43 ± 29.49 ^a^	36.55 ± 8.81 ^a^	48.42 ± 7.18 ^a^
7	*E. microcarpus*	2020.09	107.17 ± 14.68 ^a^	142.08 ± 8.40 ^a^	36.60 ± 9.04 ^a^	47.67 ± 5.70 ^a^
8	*E. microcarpus*	2020.12	105.00 ± 17.08 ^ab^	108.33 ± 16.75 ^b^	37.69 ± 5. 10 ^a^	38.86 ± 5.00 ^a^

CTR is the ratio of palisade tissue thickness to leaf thickness; SR is the ratio of spongy tissue thickness to leaf thickness; a and b represent significant differences.

**Table 4 plants-11-02449-t004:** Comparison of the chloroplast genome basic characteristics between of *E. japonicus* Beihaidao and four *Euonymus* species.

	*E. japonicus* Beihaidao	*E. hamiltonianus*	*E. phellomanus*	*E. schensianus*	*E. szechuanensis*
Total length Total length (bp)	157,661	157,360	157,543	157,702	157,465
Total GC content Total GC (%)	37.26	37.25	37.30	37.19	37.18
Length of LSC zone LSC Length (bp)	85,932	86,399	86,299	86,026	86,257
GC content in LSC area GC content in LSC region (%)	35.09	35.10	35.18	35.02	35.00
SSC zone length SSC Length (bp)	18,363	18,317	18,536	18,528	18,472
GC content in SSC area GC content in SSC region (%)	31.77	31.70	31.78	31.70	31.63
IR zone length IR Length (bp)	26,683	26,322	26,354	26,574	26,368
GC content in the IR region GC content in IR region (%)	42.65	42.72	42.71	42.62	42.70
Length of coding area Coding region length (bp)	79,512	79,968	79,005	78,069	80,871
GC content of the coding region GC content in coding region (%)	38.11	38.03	38.15	38.19	38.02
Length of non-coding area Noncoding region length (bp)	78,149	77,392	78,538	79,633	76,594
GC content of non-coding regions GC content in non-coding region (%)	36.40	36.44	36.44	36.21	36.29
Number of protein-coding genes Protein-coding gene number	87	86	85	86	89
GC content of protein-coding region GC content in protein-coding region (%)	38.11	38.03	38.15	38.19	38.02
rRNA GC content rRNA GC content (%)	55.41	55.36	55.40	55.38	55.38
tRNA GC content tRNA GC content (%)	53.39	53.13	52.99	53.10	53.32
Number of tRNAs Total tRNA	37	37	37	37	37
rRNA number Total rRNA	8	8	8	8	8
Number of genes Total gene number	132	131	130	131	134

**Table 5 plants-11-02449-t005:** Eleven regions of highly variable sequences (Pi > 0.03) of *Euonymus* species.

High Variable Marker	Length	Nucleotide Diversity
*trnH_psbA*	479	0.0827796
*trnS_trnG*	990	0.0598214
*trnG_trnR*	646	0.0419463
*petN_psbM*	581	0.0375138
*psbZ_trnG*	896	0.0727320
*trnM_atpE*; *trnV_trnM*	683	0.0477591
*rbcL_accD*	1400	0.0531015
*rpl22_rps19*	397	0.0391588
*ycf1_ndhF*	1236	0.0712147
*ndhF_rpl32*	857	0.0549155
*rpl2_rps19*	491	0.0446429

**Table 6 plants-11-02449-t006:** Nine regions of highly variable sequences (Pi > 0.12) of Celastraceae species.

High Variable Marker	Length	Nucleotide Diversity
*trnH_psbA*	581	0.1703297
*trnS_trnG*	758	0.2466063
*trnG_trnR*	102	0.1406121
*psbZ_trnG*	329	0.1709402
*trnP_ndhJ*	418	0.1515152
*psaI_ycf4*	244	0.1283296
*trnP_psaJ*	237	0.1576169
*ycf1_ndhF*	692	0.1348788
*ycf1*	420	0.1278873

**Table 7 plants-11-02449-t007:** Eleven positive selection genes in Celastraceae species.

Positive Selection Genes	Group	Length	Ka/Ks	*p*-Value
*atpE*	*E. hamiltonianus* vs. *E. japonicus* Beihaidao	399	50	0
	*E. szechuanensis* vs. *E. japonicus* Beihaidao	399	50	0
*atpI*	*E. schensianus* vs. *E. japonicus* Beihaidao	741	50	0
	*E. szechuanensis* vs. *E. japonicus* Beihaidao	741	50	0
*ndhB*	*S. amplifolia* vs. *E. japonicus* Beihaidao	1530	50	0
*ndhC*	*E. phellomanus* vs. *E. japonicus* Beihaidao	360	50	0
	*E. schensianus* vs. *E. japonicus* Beihaidao	360	50	0
	*E. szechuanensis* vs. *E. japonicus* Beihaidao	360	50	0
*petL*	*C. edulis* vs. *E. japonicus* Beihaidao	93	50	0
	*E. hamiltonianus* vs. *E. japonicus* Beihaidao	93	50	0
	*E. phellomanus* vs. *E. japonicus* Beihaidao	93	50	0
*psaI*	*E. phellomanus* vs. *E. japonicus* Beihaidao	111	50	0
*psbH*	*E. szechuanensis* vs. *E. japonicus* Beihaidao	219	50	0
*rpl20*	*E. szechuanensis* vs. *E. japonicus* Beihaidao	351	50	0
	*E. schensianus* vs. *E. japonicus* Beihaidao	351	50	0
*rpl2*	*E. schensianus* vs. *E. japonicus* Beihaidao	822	50	0
	*E. szechuanensis* vs. *E. japonicus* Beihaidao	822	50	0
*rps12*	*C. edulis* vs. *E. japonicus* Beihaidao	369	50	0
	*E. fortunei* vs. *E. japonicus* Beihaidao	369	50	0
*rps8*	*E. schensianus* vs. *E. japonicus* Beihaidao	393	50	0
	*E. szechuanensis* vs. *E. japonicus* Beihaidao	393	50	0

**Table 8 plants-11-02449-t008:** Gene function annotation of 11 positive selection genes.

Qeury Name	Length	NR Gene ID	Swiss ID	COG/KOG	Function	KO/Gene_ID	KEGG Gene Name
atpE	133	AKF33728.1	sp|Q09X11 |ATPE_MORIN	COG0355 |KOG1758	C	K02114	*ATPF1E*, *atpC*
atpI	247	AKF33710.1	sp|P69372 |ATPI_TOBAC	COG0356 |KOG4665	C	K02108	*ATPF0A*, *atpB*
ndhB	510	AKF33783.1	sp|P0CC41 |NU2C2_CARPA	COG1007 |KOG4668	C	K05573	*ndhB*
ndhC	120	AKF33727.1	sp|A7Y3E3 |NU3C_IPOPU	COG0838 |KOG4662	C	K05574	*ndhC*
rpl2	274	AVY52428.1	sp|Q4VZK5 |RK2_CUCSA	COG0090 |KOG0438	J	K02886	*RP-L2*, *MRPL2*, *rplB*
rpl20	117	AKF33744.1	sp|P06386 |RK20_TOBAC	COG0292 |KOG4707	J	K02887	*RP-L20*, *MRPL20*, *rplT*
rps8	131	AKF33757.1	sp|Q49KW3 |RR8_EUCGG	COG0096 |KOG1754	J	K02994	*RP-S8*, *rpsH*
rps12	123	ATD85454.1	sp|Q0ZIZ5 |RR12_VITVI	COG0048 |KOG1750	J	K02950	*RP-S12*, *MRPS12*, *rpsL*
petL	31	_	sp|A4QJU9 |PETL_OLIPU	_	_	_	*_*
psaI	37	AVY52374.1	sp|P56768 |PSAI_ARATH	_	_	K02696	*psaI*
psbH	73	AKF33751.1	sp|P69553 |PSBH_OENEH	_	_	K02709	*psbH*

**Table 9 plants-11-02449-t009:** Related pathway analysis of 11 positive selection genes.

Path Way	Layer 1	Layer 2	Pathway Definition	Kos List
ko00190	1. Metabolism	1.2 Energy metabolism	Oxidative phosphorylation	*atpE*: K02114; *atpI*: K02108; *ndhB*: K05573; *ndhC*: K05574
ko00195	1. Metabolism	1.2 Energy metabolism	Photosynthesis	*atpE*: K02114; *atpI*: K02108; *psaI*: K02696; *psbH*: K02709
ko01100	1. Metabolism	1.0 Global and overview maps	Metabolic pathways	*atpE*: K02114; *atpI*: K02108; *ndhB*: K05573; *ndhC*: K05574
ko03010	2. Genetic Information Processing	2.2 Translation	Ribosome	*rpl20*: K02887; *rpl2*: K02886; *rps12*: K02950; *rps8*: K02994

## Data Availability

The original contributions presented in the study are not publicly available at present.

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
