# Peer review of "Cold Resistance of Euonymus japonicus Beihaidao Leaves and Its Chloroplast Genome Structure and Comparison with Celastraceae Species"

_plants, 2022, doi:10.3390/plants11192449_

Round 1
Reviewer 1 Report
Cai et al., submitted the manuscript entitled “Cold Resistance of Euonymus japonicus Beihaidao Leaves and its Chloroplast Genome Structure and Comparison with Celastraceae Species “for publication consideration in Plants.
This study aimed to provide information a preliminary study 14 of the cold resistance of this species, evaluated its performance during winter, assembled and anno- 15 tated its chloroplast genome and performed a series of analyzes to investigate its gene structure GC 16 content, sequence alignment, and nucleic acid diversity. The authors reported the phylogenetic relationships of 37 species inferred that members of the Euonymus genus do not form a clade and that E. japonicus Beihaidao is closely related to E. japonicus and E. fortunei. functional positive selected genes were identified, which may have played an important role in the process of Celastraceae species adapting to environmental changes
The scientific soundness of this manuscript is acceptable. It meets the aims and scope of plants journal.
This study is interesting. The manuscript is clearly presented, however, there is clear shortcomings in sections of Method and Materials and Results beside of English issue.
Some issues required attention:
The typo occurred, for example, please carefully double check spelling and grammer through whole manuscript, typos occurred, for example
In Line 24: traceae species, the variation in the chloroplast genome sequencewas lower, and the gene structure
In Figure 3. Changes in leaf surface color difference in three Euonymus species. Description of sampling time should not use “months”. “September 2020, December, 2020 and May, 2021” should be used directly.

Author Response
Response to Reviewer 1 Comments
Dear Reviewer:
Thank you for your comments on our manuscript entitled “Cold Resistance of Euonymus japonicus Beihaidao Leaves and its Chloroplast Genome Structure and Comparison with Celastraceae Species” (ID: plants-1873061). All of those comments are valuable and very helpful for revising and improving our paper, as well as providing crucial direction for our research. We have studied the comments carefully and have made correction which we hope meet with approval. And the revised version is marked by “Track Changes” function. The main corrections in the paper and responds to the comments are as following:
Point 1: The typo occurred, for example, please carefully double check spelling and grammer through whole manuscript, typos occurred, for example
In Line 24: traceae species, the variation in the chloroplast genome sequencewas lower, and the gene structure
In Figure 3. Changes in leaf surface color difference in three Euonymus species. Description of sampling time should not use “months”. “September 2020, December, 2020 and May, 2021” should be used directly
Response 1: We are very sorry for the errors in our writing. We have carefully gone over the spelling and grammar, corrected improper words, and fixed mistakes in figures.
Point 2: Objective of this study seems to be focused on research E. japonicus Beihaidao’s leaves cold resistance, some basic observations have been performed, no further studies on underlined mechanisms was mentioned.
Response 2: We have revised the section describing the genes responsible for the cold tolerance of E. japonicus Beihaidao’s leaves according to the Reviewer’s comments. We have made gene function annotation and analyzed related pathway of the positive selection genes. In addition to these, we also detected 4 of the positive selection genes in existing transcriptome from other lab members, performed KEGG enrichment analysis and analyse the expression of these genes in different periods.
Point 3: It is not easy to connect the Chloroplast Genome Structure and Comparison work to cold resistance part, the manuscript looks very scattered.
Response 3: Considering the Reviewer’s suggestion, we have modified the text logic to emphasize evolutionary and genetic analysis as two objectives.
Once again, thank you very much for your comments and suggestions!

Reviewer 2 Report
The authors did a lot of analyses for freezing resistance and analysed the chloroplast genome of the variety E. japonicus Beihaidao.
I have some remarks and recommendations for the authors:
Table 1: there are some genera not mentioned with full name but only with the first letter. Please, give the full genus once per species.
In chapter 2.1 three species (variety) are mentioned: E. japonicus Beihaidao, E. japonicus and E. microcarpus. In the result section 3.1 and Table 2 E. oblongifolius is mentioned instead of E. japonicus.
The name of 3.1.1: “Repeat sequence analysis” doesn’t fit to the content of the chapter.
Chapter 3.3.1, line 398: SSC region 18363? (not 8363)
Figure 7 is of very bad quality – not readable!
Line 448, 449: “suggesting that single nucleotide repeats may play a more important role in genetic variation than other types of SSRs”. That is known for all plants, so, not necessary to mention it.
The authors did a lot of analyses with the chloroplast sequences. But, I am not sure if everything is necessary to be mentioned: e.g. Figure 8 and Figure 11. I don’t see an additional value making these analyses. And, for such a lot of analyses the interpretation (discussion) is rather short and not very deep rooted. Maybe here some more discussion of the identified genes, maybe related with cold stress, can be provided? In the abstract “adapting genes to environmental changes” are promised, but, unfortunately, climate change is not a matter of discussion later on.
Author Response
Response to Reviewer 2 Comments
Dear Reviewer:
Thank you for your comments on our manuscript entitled “Cold Resistance of Euonymus japonicus Beihaidao Leaves and its Chloroplast Genome Structure and Comparison with Celastraceae Species” (ID: plants-1873061). All of those comments are valuable and very helpful for revising and improving our paper, as well as providing crucial direction for our research. We have studied the comments carefully and have tried our best to improve the manuscript which we hope meet with approval. And the revised version is marked by “Track Changes” function. The main corrections in the paper and responds to the comments are as following:
Point 1: Table 1: there are some genera not mentioned with full name but only with the first letter. Please, give the full genus once per species.
Response 1: We are very sorry for our negligence of this problem. And we have checked and correct the Latin names.
Point 2: In chapter 2.1 three species (variety) are mentioned: E. japonicus Beihaidao, E. japonicus and E. microcarpus. In the result section 3.1 and Table 2 E. oblongifolius is mentioned instead of E. japonicus.
Response 2: We are very sorry for us confusion of these two species and it is rectified in section 2.1.
Point 3: The name of 3.1.1: “Repeat sequence analysis” doesn’t fit to the content of the chapter.
Response 3: We are very sorry for our incorrect writing, and it is corrected at line 267.
Point 4: Chapter 3.3.1, line 398: SSC region 18363? (not 8363)
Response 4: We are very sorry for our negligence of this mistake and we have corrected the data in chapter 3.3.1.
Point 5: Figure 7 is of very bad quality – not readable!
Response 5: We are very sorry for this question and the figure is replaced with a high-quality version.
Point 6: Line 448, 449: “suggesting that single nucleotide repeats may play a more important role in genetic variation than other types of SSRs”. That is known for all plants, so, not necessary to mention it.
Response 6: It is really true as Reviewer suggested that this part is not necessary to be mentioned. We have removed the unnecessary sentences according to the comments.
Point 7: The authors did a lot of analyses with the chloroplast sequences. But, I am not sure if everything is necessary to be mentioned: e.g. Figure 8 and Figure 11. I don’t see an additional value making these analyses. And, for such a lot of analyses the interpretation (discussion) is rather short and not very deep rooted. Maybe here some more discussion of the identified genes, maybe related with cold stress, can be provided? In the abstract “adapting genes to environmental changes” are promised, but, unfortunately, climate change is not a matter of discussion later on.
Response 7: We have made correction according to the reviewer’s comments. And we have revised the section describing the genes responsible for the cold tolerance of E. japonicus Beihaidao’s leaves according to the Reviewer’s comments. We have made gene function annotation and analyzed related pathway of the positive selection genes. In addition to these, we also detected 4 of the positive selection genes in existing transcriptome from other lab members, performed KEGG enrichment analysis and analyze the expression of these genes in different periods.
Once again, thank you very much for your comments and suggestions!

Round 2
Reviewer 1 Report
Revised version showed improvements. I would like to recommend for publication in Plants.
Author Response
Response to Reviewer 1 Comments
Point 1: Revised version showed improvements. I would like to recommend for publication in Plants.
Response 1: We appreciate for Reviewer’s warm work earnestly, and we still checked the spelling and correct mistake words.
Once again,thank you very much for your commnts and suggestions!

Reviewer 2 Report
The authors did nearly all of the recommended changes - thanks for that!
Only in Table 1 the full names of several genera are still missing.
Author Response
Response to Reviewer 2 Comments
Point 1: Only in Table 1 the full names of several genera are still missing.
Response 1: We are grateful to Reviewer for reviewing the paper so carefully. Considering the Reviewer’s suggestion, we have filled up all the full names in Table 1 for easier reading.
Once again,thank you very much for your comments and sugestion!
